# Load Testing of Cultural Heritage Structures and Sculptures: Unconventional Methods for Assessing Safety

**Miloš Drdácký *** and Shota Urushadze

Institute of Theoretical and Applied Mechanics of the Czech Academy of Sciences, 190 00 Prague, Czech Republic; urushadze@itam.cas.cz

*   Correspondence: drdacky@itam.cas.cz

**Abstract:** The paper presents the results of static and dynamic experimental tests conducted on historical heritage structures and sculptures. In recent years, there was an increasing need to address the behaviour of these types of structures due to several reasons. Diagnosing the actual condition of a historical structure involves not only identifying the cause of a detected defect, but also determining its impact and assessing whether the structure can continue to perform safely. This article utilises unconventional loading methods to generate measurable mechanical responses. In one case, a lifting procedure is employed to study strains in a composite structure, while in another example, the mass and movement of people are used as a form of loading. Proof load tests conducted on a monumental sculpture allowed for the determination of load distribution among its heterogeneous structural components, namely a bronze shell and an iron reinforcing frame. Furthermore, the static and dynamic loading of a ceiling supported by a masonry vault demonstrated its ability to withstand anticipated loads resulting from unconventional use as a temporary exhibition space.

**Keywords:** proof load test; ambient vibration; historic structure; frequency analysis





## 1. Introduction

Assessment of cultural heritage structures and sculptures from the point of view of their health and capacity to sustain acting loads represents one of the most difficult engineering tasks due to several reasons. In particular, it is difficult to create computational models of complex object shapes and unknown internal geometry. Furthermore, knowledge of the material characteristics of structures is usually insufficient. Moreover, the objects may be composed of multiple materials that can interact with each other. Therefore, even qualitative data on the behaviour of heritage structures and sculptures is useful and acceptable for an assessment of the mechanical response of cultural heritage structures and sculptures supporting decisions concerning their management or interventions' planning.

In situ proof load tests are commonly used effective methods of assessing the condition and safety of existing structures [1,2]; for example, historic bridges or ceilings [3,4], roofs, and frames [5], especially in earthquake-prone situations [6]. Proof load tests are necessary also for testing of replicas of historic timber machines, such as medieval cranes, before they are introduced into service on construction sites [7]. In situ tests provide the engineers with data necessary for the redesign or safety assessment of existing buildings, which is mostly used for masonry, timber, or concrete structures [8–10]. They were applied also for study of historic staircase structures [11,12] or free-standing gables [13]. In such cases, the structural response can be usually investigated with rather simple static or dynamic models and the tests provide exact data. However, it is typical for historical constructions and objects of art that they exist in a very wide range of variants, and therefore, the methods of examining their behaviour require a wide range of appropriate approaches. The present article shows two examples that demonstrate procedures based on a creative application of non-traditional load test methods.

Sculptors use rigid frameworks or skeletons to support plastic material during modelling as well as during assembly of large sculptures composed of sections. Therefore, armatures or structural frames are frequently used to sustain or reinforce 3D art objects [14], which requires the close cooperation between artists and engineers. Large representative sculptures meant for outdoor display are typically fashioned of bronze or other types of sheet metal, and they require armatures for internal support and stability of the shape. There are well-known examples of the work of excellent engineers who designed frames supporting large sculptures; for example, a large steel armature designed by Gustave Eiffel holds up the Statue of Liberty in New York [15]. Similarly, a reinforced concrete frame designed by Bedřich Hacar supports the third largest equestrian statue in the world, depicting the Hussite military leader Jan Žižka in Prague [16] (Figure 1). Such frames are not considered to interact with the sculpture and are designed to carry the whole mass of the art object.

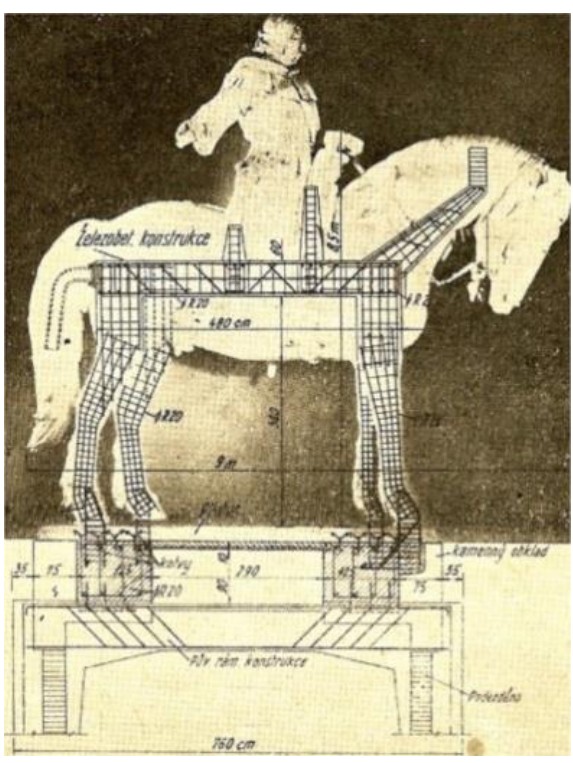 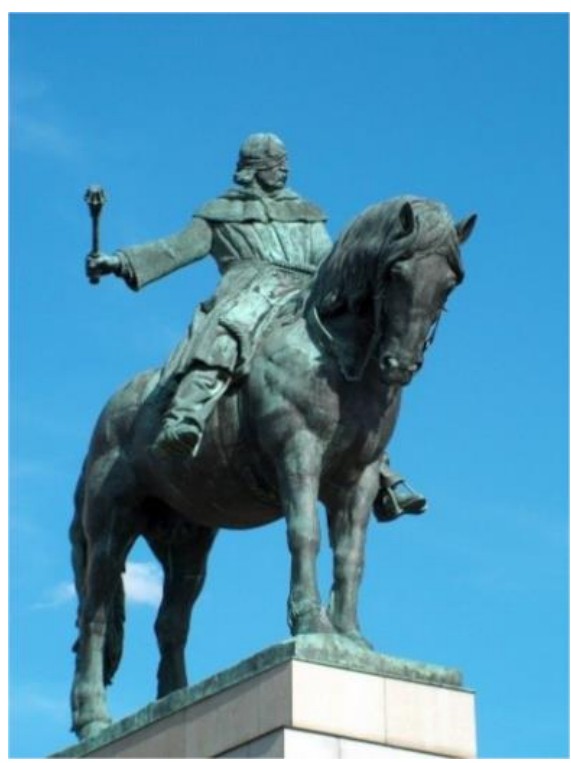

**Figure 1.** Bronze equestrian statue depicting the Hussite hetman Jan Žižka in Prague (**right**) and its reinforced concrete supporting frame (**left**).

However, the bronze sculptures are shells with a complex geometry, and their rigidity and load-bearing capacity are not negligible. Taking into account these characteristics, some small and medium-size sculptures are not supported with frames, which carry the total mass of the sculpture. Their armatures only contribute to the overall stability of the sculpture, even though they were mostly designed as scaffoldings facilitating the assembly of the sculpture composed of precast elements. In such a case, the bronze shell and the armature create a composed structure and share their load carrying behaviour.

The problem of interaction of structural elements made of different materials occurs not only in the above-mentioned example. It is further typical in composed systems when steel structures are combined with timber elements, e.g., steel roof girders interacting with timber purlins stabilised with timber boards. The task to determine the degree of interaction of individual parts in a complex system was reliably studied only experimentally due to uncertainties in connections and the force transition between individual parts, which is difficult to assess. During the restoration of the triga sculpture on the roof of the National Theatre in Prague, it was possible to investigate the extent of the interaction of the bronze

shell of the statues and the internal reinforcing steel rod frame in the transfer of the load acting on the statues.

It is similarly difficult to assess the response of vaulted ceilings to static or dynamic loads. This is a frequent problem in the public spaces of museums, galleries, or castles, where a larger number of people can enter. If we know the geometry of the vault and its material properties, we can perform a calculation of the stress in the vault [17] and assess its bearing capacity, and if necessary, verify the result by a load test. In many cases, however, it is sufficient to perform a load test and monitor the course of deformations. Such verification is faster and cheaper. This was a case of planning an exhibition in a small renaissance chateau with a concentration of visitors loading a large-span subtle brick vault.

Concerning the dynamic behaviour of historic structures, extensive research was carried out by the UK Transport and Road Research Laboratory [18], where building damage resulting from vibrations generated by traffic is studied. There are often cases where it is difficult to determine the level of vibration caused by traffic that can damage a structure, and the measured values do not clearly reveal their cause and danger [19]. The level of vibration significantly depends on the soil type and stratification; for example, in soil with reduced stiffness or dampened soil, vibration levels increase. In areas underlain by a soft silty clay with a depth of 7 to 15 m, vibrations induced by traffic reach even greater values [20]. In literature, there are known facilities dealing with the effects of technical seismicity, and numerous applications of modal analysis of civil structures such as bridges and tall buildings, but only exceptionally the effects on historical buildings that have a complex structure. An analysis of the dynamic effects on different types of historic buildings and their constructions can be found [21–26].

In situ proof load tests are further used to determine the natural frequencies, vibration shapes, and damping of historic structures subjected to the threats of earthquakes [27]. Another common reason for load testing is to assess the safety of structures and buildings after partial damage [28].

## 2. Load Tests on Bronze Sculptures

The triga sculptures designed by Czech Sculptor Bohuslav Schnirch were placed on the pylons of the roof of the National Theatre in Prague in 1911 (Figure 2) and were restored three times since then. In 1940, when the statues were opened, they were found to be in surprisingly poor condition. According to the restoration report, the lower parts of the horses' legs were filled in with concrete in order to "increase stability, due to the possible bombing of the city and thus the shocks or onslaught of blast waves". It is not known by what methods the deformation response and possible stresses in the structure were verified under these dynamic effects. The statues of the horses have relatively little rigidity in the horizontal direction and perpendicular to the median plane of the horses' bodies, which may have raised concerns about their stability. During further restoration in 1966 and 1981, the construction of the horses was not interfered with. Several approximate static calculations were made—most recently, during the survey described in this paper. It was found that the stress of the inner reinforcing steel frame assuming the full load of the weight of the statue is so high in some cross-sections that this structure would not be able to safely transfer the load of the statues without the cooperation of the bronze shell. It was expected that the bronze shell itself is sufficiently rigid and strong to accommodate a substantial part of its own weight and external loads.

### 2.1. Structural and Material Data

The study was limited to only one horse from the western triga sculpture. The studied statue of the horse acts statically as a console of complex shape, fixed to the roof structure by three supports: two hind legs and a tail. The body, head, and front legs protrude freely into space. A general view of the horses is shown in Figure 3. The legs and tail are made up of hollow thin-walled bronze profiles of irregular cross-section and irregular thickness,

and the body is statically a rather short thin-walled hollow rod than a shell. The spatial rigidity of the bronze skin is considerable.

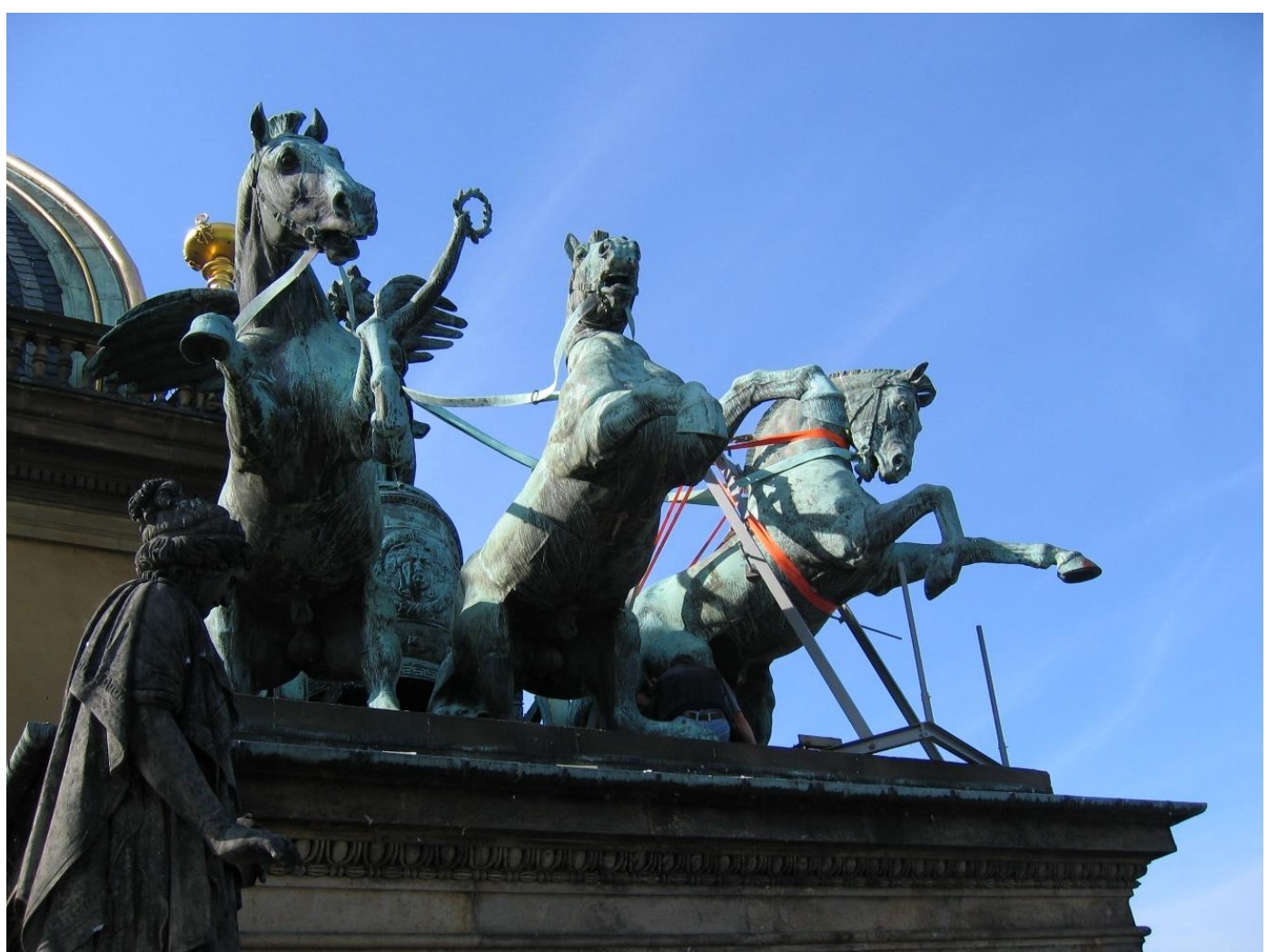

**Figure 2.** Western bronze triga horses with carriage on the pylons of the roof of the National Theatre in Prague.

The statue of the horse is supported by a steel skeleton of complex shape, made of solid profiles of square cross-sections of 50 mm × 50 mm that are forge-shaped according to the needs of the geometry of the statue. This steel frame is firmly connected to the steel structure of the base grid built into the pylon roof slab. The bronze legs and tail are attached to the steel skeleton, sometimes in very close contact. A diagram of the skeleton is shown in Figure 3, where the interior steel skeleton is marked in red.

According to the results of tests on samples, the reinforcing frame is made of steel produced in a Bessemer converter, Table 1. It has a marked yield strength (about 350 to 360 MPa), an average tensile strength of 436 MPa and a ductility of 23.8%. Its modulus of elasticity is around 98 GPa. The bronze shell is made of a rather poor quality material with a number of defects and a relatively low strength of about 115 MPa, with an average modulus of elasticity of about 46 GPa, Table 2 (see the following tables of material characteristics based on tests carried out at the Institute of Metallic Materials and Corrosion Engineering of the Institute of Chemical Technology in Prague).

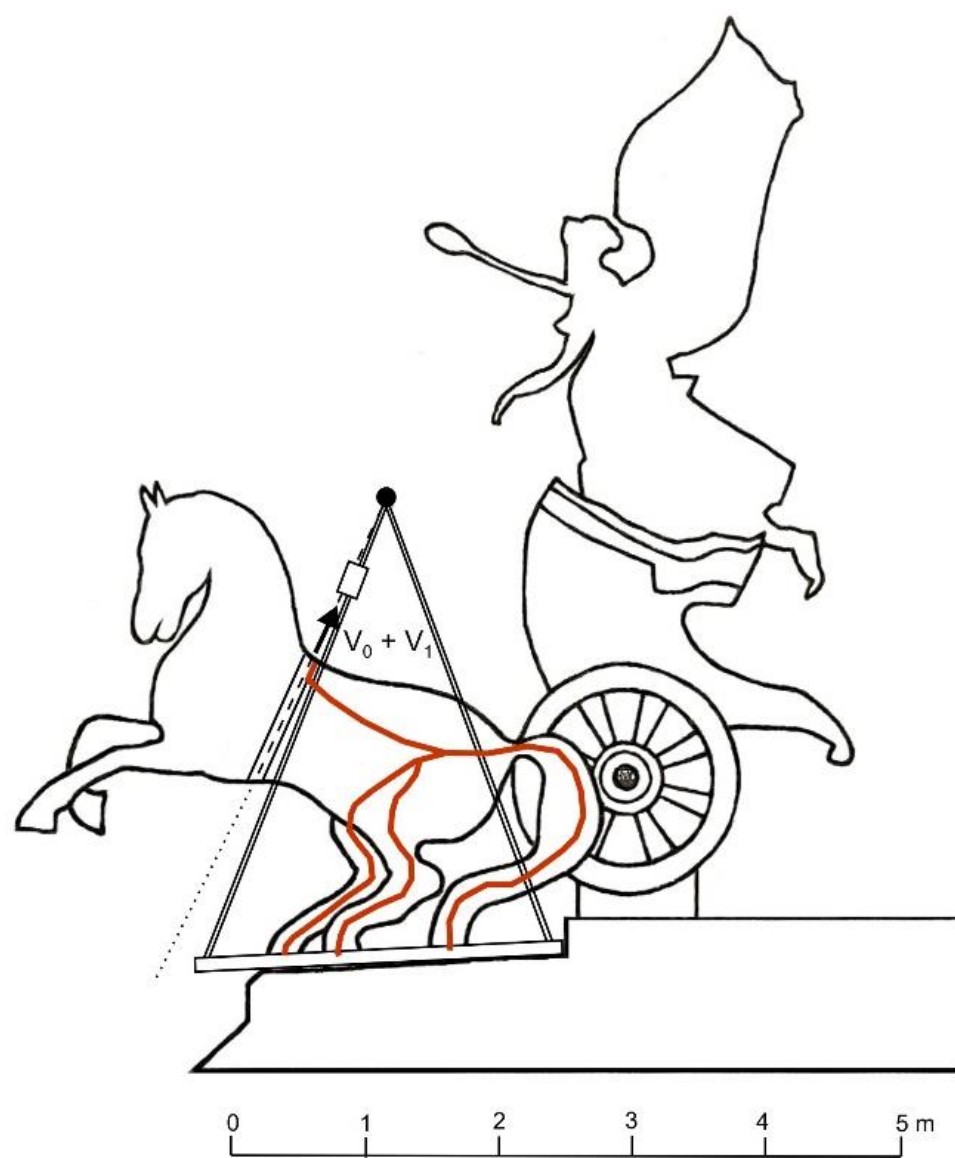

**Figure 3.** The figure shows a schematic view of the inner steel skeleton (red) with a temporary triangular supporting frame and the position of load cells.

**Table 1.** Material characteristics of steel.

| | Specimen | Ductility $A_{5.65}$ [%] | Strength in Tension $R_m$ [MPa] | Upper Yield Strength $R_{eH}$ [MPa] | Lower Yield Strength $R_{eL}$ [MPa] | Modululus of Elasticity in Tension $E$ [GPa] |
|---|---|---|---|---|---|---|
| | 1 | 17.5 | 422 | 363 | 338 | 83 |
| | 2 | 26.5 | 465 | 382 | 375 | 102 |
| **steel** | 3 | 27.5 | 422 | 338 | 335 | 108 |
| | Average | 23.8 | 436 | 361 | 349 | 98 |
| | $\sigma_{n-1}$ | 5.5 | 25 | 22 | 22 | 13 |

**Table 2.** Material characteristics of bronze.

| Specimen | | Ductility $A_{5.65}$ [%] | Strength in Tension $R_m$ [MPa] | Yield Strength $R_{p0.2}$ [MPa] | Modululus of Elasticity in Tension E [GPa] |
|---|---|---|---|---|---|
| bronze | TGZB1 | 23.4 | 240 | 121 | 49 |
| | TGZB2 * | 8.5 | 159 | 108 | 43 |
| | TGVB1 | 1.5 | 205 | n.a. | n.a. |
| | TGVB2 * | 0.4 | 152 | n.a. | n.a. |
| | TGVB3 * | 0.4 | 164 | n.a. | n.a. |

star indicates test specimens with a significant defect.

### 2.2. In Situ Static Load Test Arrangement

Due to a number of objective reasons (high rigidity of the statue in the vertical direction, lack of space for placing an anchoring, a heavy load frame, etc.), it was not possible to perform a classic load test arrangement and it was necessary to find another suitable way of inferring force changes in the structure. We produced the load by lifting the horse, which advantageously used the suspension structure already built for the restoration purpose (see Figure 3) and was acceptable for assessing the ratio of the interaction of the steel frame and the bronze casing.

Load cells for measuring the applied force were inserted between the suspension straps and the existing steel suspension structure (frame). Due to the small working height between the statue and the frame, two load cells (marked V0 and V1 in the pictures) were used to measure the force in the straps on both sides of the horse, Figure 4A. The load cells were suspended in the system after completely loosening the straps applied for the restoration process and removing them in the parts where the force was not measured (neck and tail). Deformation and proportional deformation were monitored on the statue during the release of the straps and later during loading. The straps were tensioned by a tooth mechanism, which did not allow for equalisation of the forces on both sides of the horse and made the measurement more difficult. Therefore, it was not possible to achieve a completely symmetrical stress in both straps, nor a continuous increase in force. One material test systems ("MTS") load cell with a capacity of 25 kN and one "LUKAS" load cell with a capacity of 50 kN were used for the measurement. The structure was loaded continuously, and the force and other monitored values (displacement and proportional strain) were continuously measured.

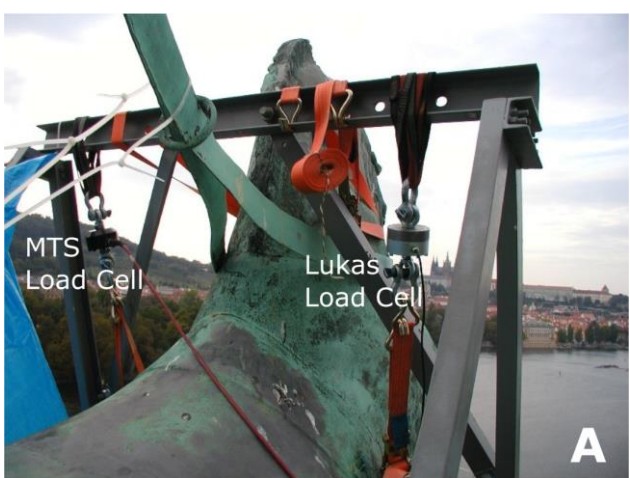
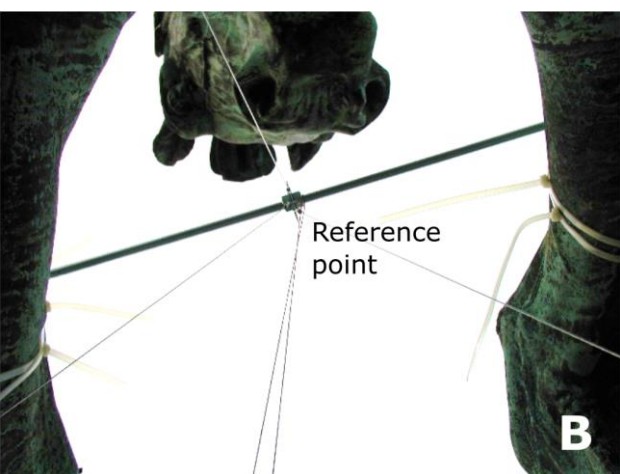

**Figure 4.** Load cells for measuring the applied force (**A**); the centre of the connection between the front legs was the reference point for the displacement measurements (**B**).

The displacement of the statue complex cantilever end required the creation of a reference point. This point was considered the centre of the connection between the front legs, fixed by means of a threaded rod attached to the statue (Figure 4B). The spatial record of the displacement characterises the horse's deformation under load and was determined by measurements using three rotary potentiometric path sensors. The individual pickups were marked PP (on the right when facing North), PL (left), and PZ (to the rear below the horse's chest).

The relative strain was measured by means of electrical resistance strain gauges glued to the examined statue, its bronze shell, and to the reinforcing steel structure. An example of the location and marking of strain gauges on the steel structure is illustrated in Figure 5. Here, H denotes the dorsal profile and PV right leg reinforcing bar with sides facing north (S) or south (J) sides. The measurements were performed on the dorsal profile (strain gauges CH 120, 121, 122, and 123) and on the right (strain gauges 127, 128) and left (strain gauges 125, 126) sides of the back leg (CH denotes measuring and recording channel). On the bronze coat, proportional deformation was measured on both hind legs in two profiles (above the hoof and on the thigh) and on the tail, as shown in Figure 6. Strain gauges were placed in the vertical plane on the upper and lower surface of the hollow profile of the foot, in the horizontal plane in the assumed level of the neutral axis. This arrangement enables calculation of the bending moments and axial forces in the measured element. Strain gauges on the right (and left) leg were marked PN (and LN), H, and D for the upper and lower surfaces; in the neutral axis PN (and LN), P, and L according to the right and left sides; the numbers of the corresponding measuring channels are shown in the figures. Strain gauges on the tail are marked O-H (and D, P, and L) according to the same scheme. The overall arrangement of strain gauges can be seen in Figure 7.

### 2.3. Dynamic Response Measurements

The aim of the dynamic measurement was to demonstrate the possible influence of ambient technical seismicity, mainly from automobile and rail transport at the level of the triga anchorage. The dynamic response was measured using three pairs of Wilcoxon CMSS 916 VD sensors to determine speed and vibration deflection values. An ENDE-VCO 86 accelerometer was used to detect the acceleration response. Each pair of sensors recorded vertical and horizontal responses. The recordings were continuously monitored and recorded for further analysis on a portable computer. Signal processing was performed using MATLAB.

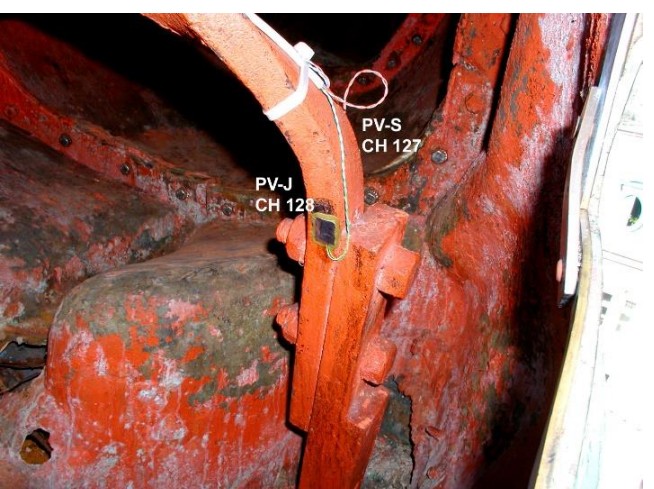 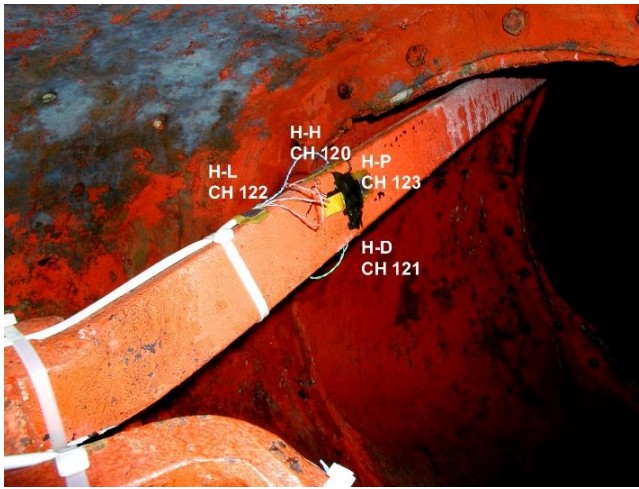

**Figure 5.** Strain gauges on the right hind leg upright (**left**) and dorsal profiles (**right**).

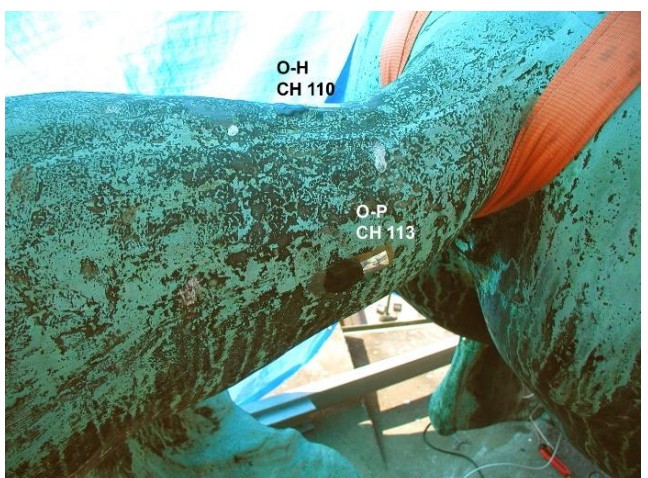 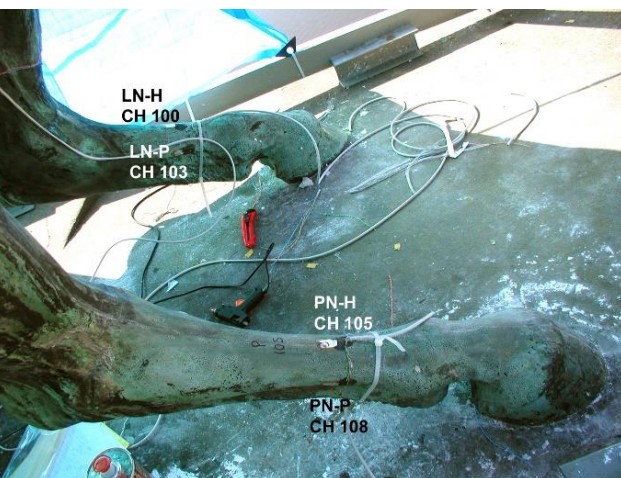

**Figure 6.** Strain gauges on the bronze shell; on the tail (**left**) and on the hind legs (**right**).

In order to capture the oscillation on the pylon of the roof of the National Theatre and on the statue at the same time, three measuring points were selected. An overall view of the locations of all the sensors is shown in Figure 8. The dynamic response was measured both during standard road traffic and when the statue vibrated due to an impulse in the horizontal or vertical direction.

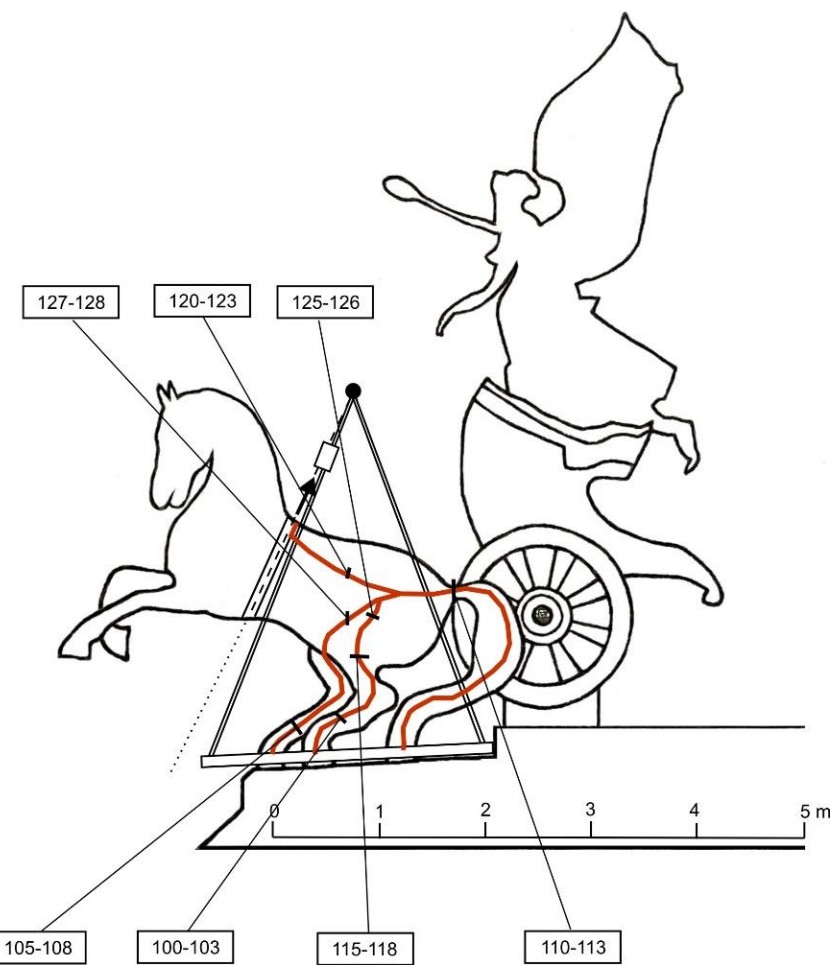

**Figure 7.** Overall arrangement of the groups of strain gauges on the sculpture.

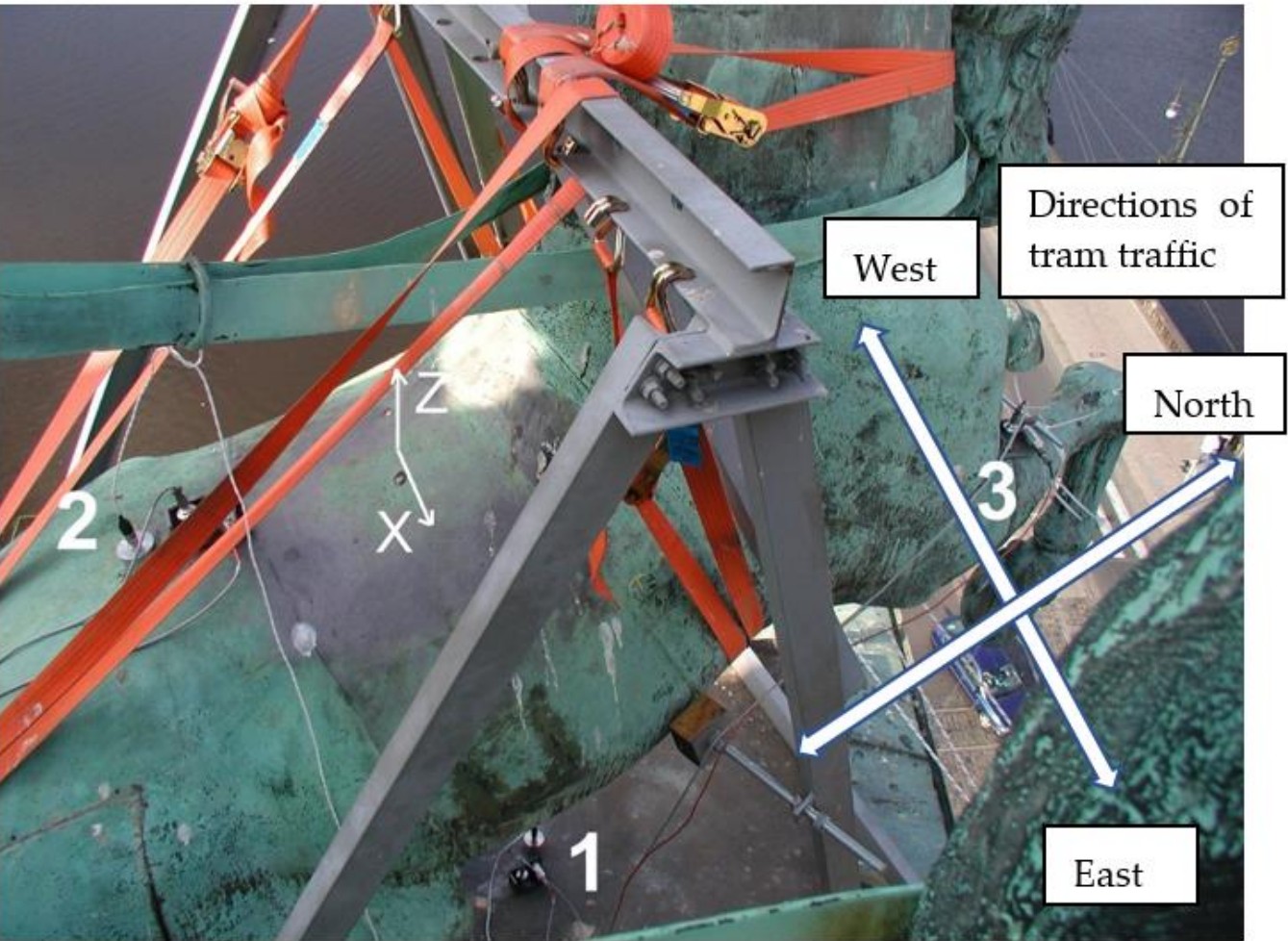

**Figure 8.** Locations of vibration sensors (1, 2, 3) and the directions of tram movement.

*2.4. Static Test Results and Discussion*

The deflection of the statue was calculated from the measured values and converted into three mutually perpendicular components: vertical deflection (z), horizontal deflection in the direction of the horse's axis (y), and horizontal deflection in the direction perpendicular to the horse's axis (x). The results of the displacement values calculated in this way are shown in Figure 9. Proportional strains on the bronze shell and on the reinforcing frame were measured by electrical resistance strain gauges, and from these measurements, the stresses on the surface of the structure under the selected loading method were calculated. Result examples are shown in Figure 10.

From the measured results, it is clear that the bronze shell significantly contributes to the transfer of the statue's own weight to the support. During unloading, higher stresses were measured in the shell. The cross-sectional areas of the profiles are for the most part larger in the shell than in the steel frame, except for the profile above the hoof where the bronze tightly encircles the steel profile. Without detailed knowledge of the cross-sectional areas at a measurement point, it is difficult to infer the specific contribution of interaction. According to the behaviour of the structure, we estimate that the bronze cladding can transfer about 75% of the statue's own weight to the support in the mostly pressed and bent elements (the legs).

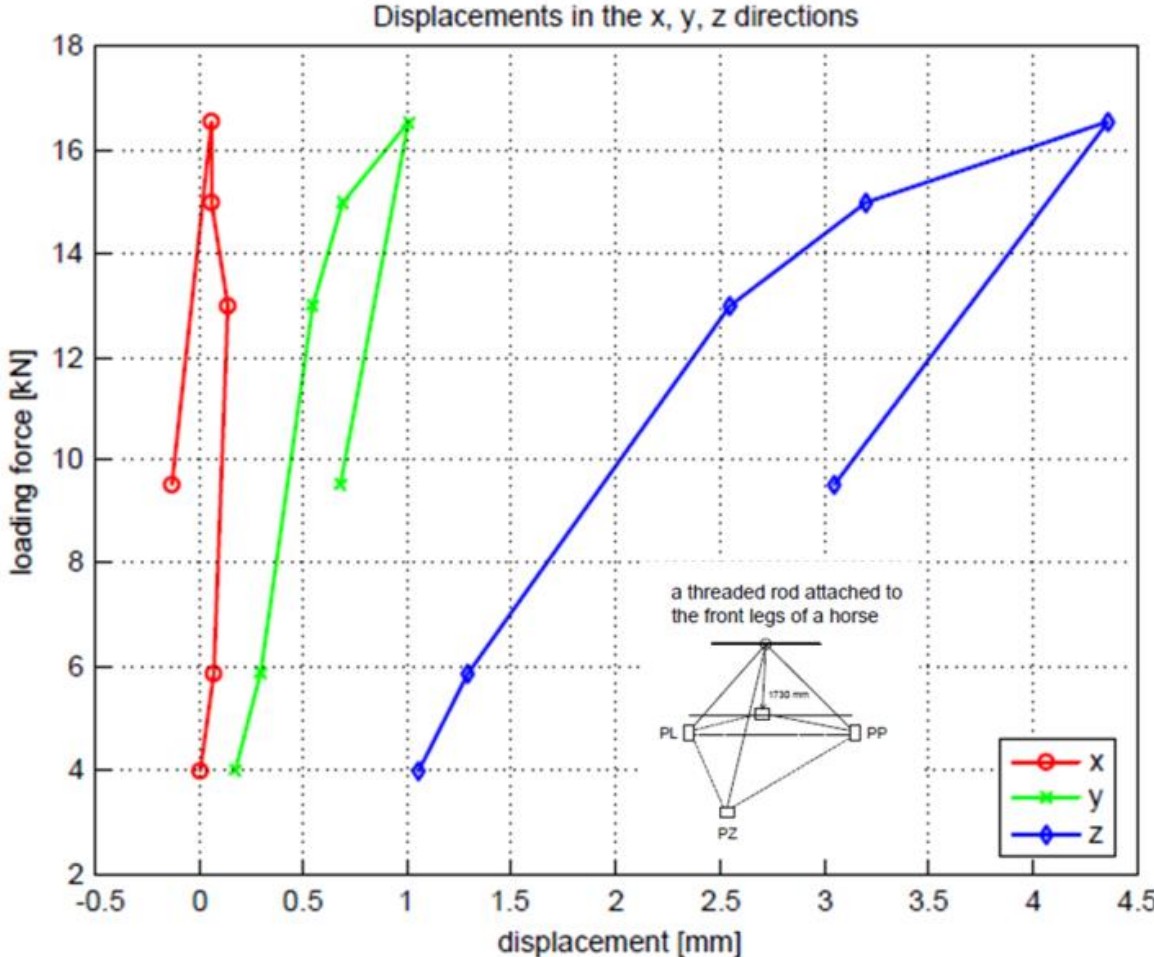

**Figure 9.** Translation of a fixed point between the horse's front legs (the displacements in the x, y direction are horizontal; the vertical deflection is measured in the z direction).

*2.5. Dynamic Response and Comments*

In total, twenty-nine representative records were measured for the vibration analysis. The oscillation time course for each of the 40 s-long recordings was analysed and a short time interval was selected from each for better representation. An overview of the selected forced frequencies of dynamic response to tram and car passages is shown in Figure 11.

The frequency components of the time record from the massive building where the statue is located are shown in the pictures above. The frequency composition in the vertical and horizontal directions corresponds to the wide band spectrum typical for dynamic loading of the induced traffic. The figures in the middle and bottom correspond to the frequency of the statue in the vertical and horizontal directions. In these cases, the spectrum peaks show the natural frequency of the statue, which can be amplified at the resonance excitation frequencies.

WILCOXON sensors were set to a sensitivity of 141 mV/mms$^{-1}$ for velocity and 13.5 V/mm for deflection. The ENDEVCO sensor sensitivity was set to 1000 mV/mms$^{-2}$ for acceleration. The obtained electrical signals can be converted to values for acceleration, velocity, and deflection. Two oscillation directions were observed: horizontal and vertical. The record files are summarised in Table 3.

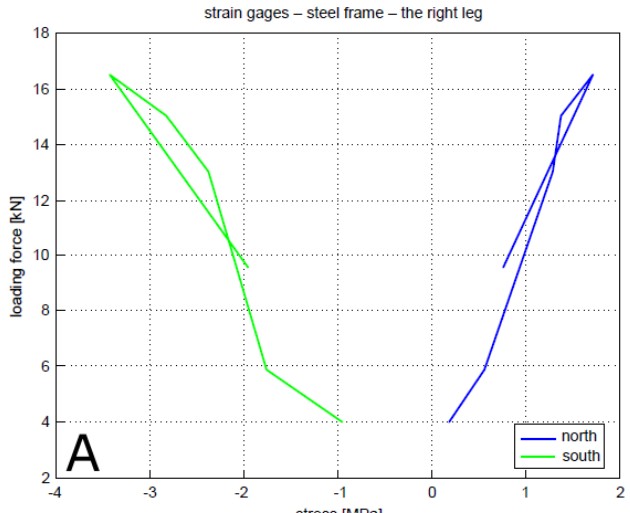
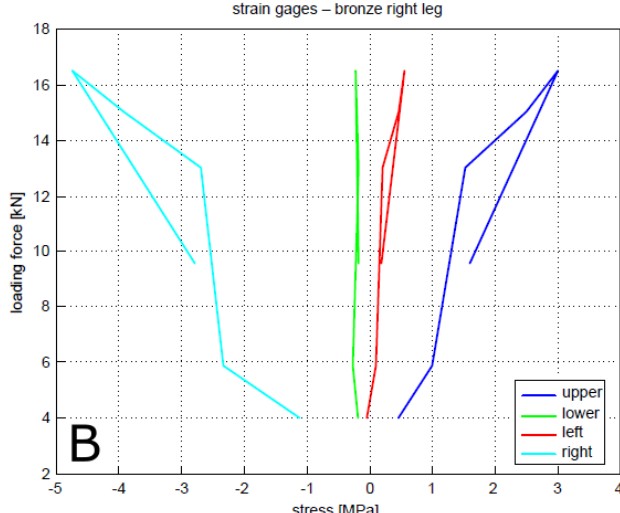

**Figure 10.** Stresses in the right-hand post of the reinforcing steel frame (**A** see Figure 5) and stresses in the bronze profile of the right leg above the hoof (**B** see Figure 6) during the lifting and the lowering of the sculpture.

**Table 3.** Excitation.

| File Nr. | Street Traffic Excitation | File Nr. | Street Traffic Excitation |
|---|---|---|---|
| 1 | Tram W-E | 16 | No traffic. Wind 2–4 ms⁻¹. |
| 2 | Tram E-W | 17 | Tram E-W (one wagon). Tram W-E follows. |
| 3 | Tram W-E at the beginning of recording. Tram N-E at the end of recording. | 18 | Tram along the river N-S. |
| 4 | Tram W-E at the beginning of recording. Tram E-W follows. | 19 | Tram W-E. |
| 5 | Tram E-N | 20 | Tram W-E at the beginning of recording. Tram along the river S-N follows. |
| 6 | Tram E-W in the middle of recording. Tram E-N at the end of recording | 21 | Tram W-E. |
| 7 | Tram W-E | 22 | Tram W-E at the beginning of recording. Tram E-W follows. |
| 8 | Tram W-E at the beginning of recording. Tram E-W follows. | 23 | No traffic. |
| 9 | Tram W-E | 24 | No traffic. |
| 10 | No traffic. Tram E-W at the end of recording | 25 | Cyclic manual excitation in horizontal direction. |
| 11 | Tram W-E at the beginning of recording. Tram E-W follows. | | Shock loading |
| 12 | Tram along the river S-N. | 26 | Shock load in horizontal direction. |
| 13 | Tram E-W in the middle of recording. Tram W-E follows. | 27 | Shock load in horizontal direction. |
| 14 | No traffic. Tram W-E at the end of recording. | 28 | Shock load in vertical direction. |
| 15 | No traffic. Wind 2–4 ms⁻¹. | 29 | Shock load in vertical direction. |

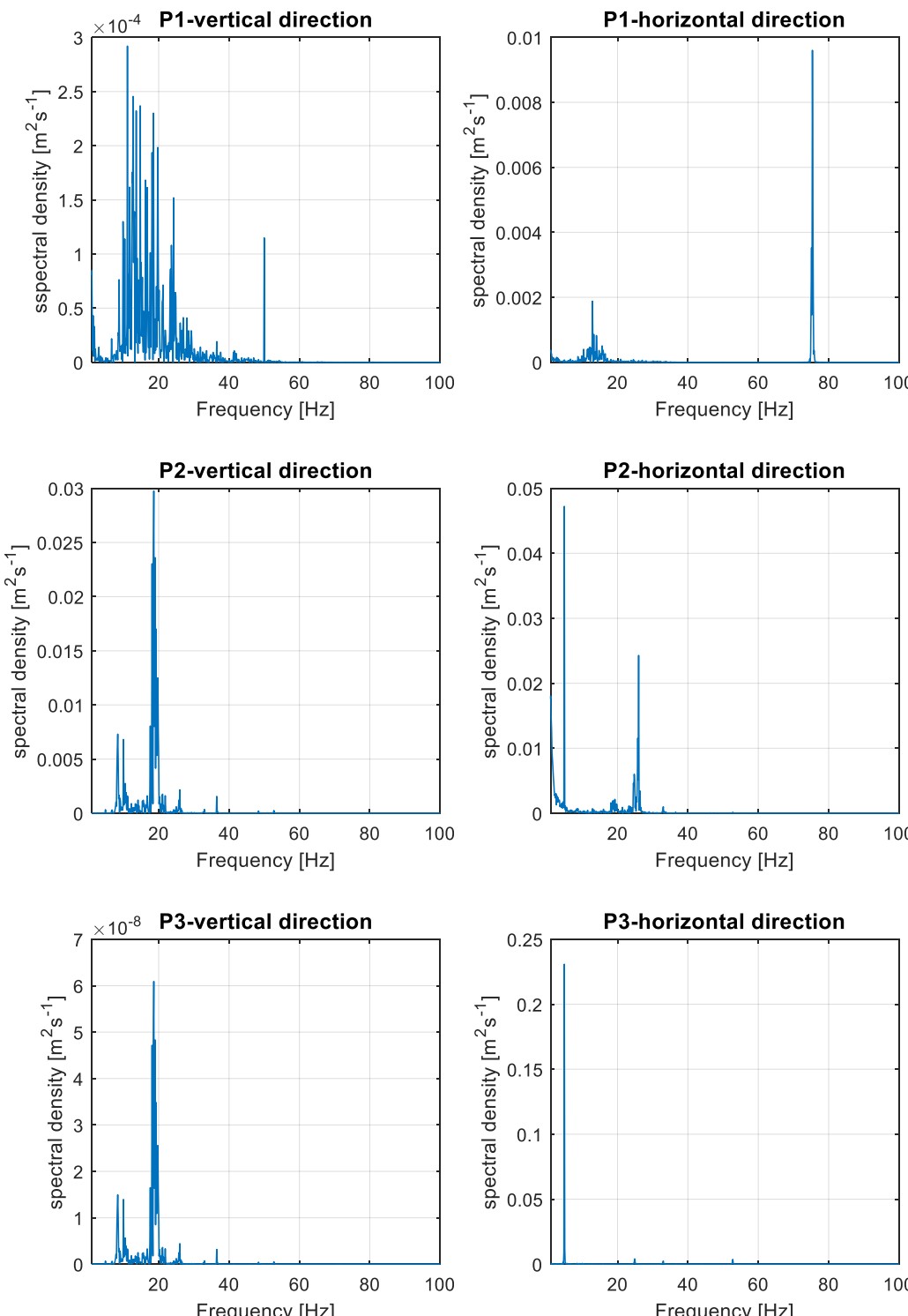

**Figure 11.** Record example for the vertical and horizontal directions at points 1, 2, and 3: a tram passes travelling east and another goes along the river moving south.

From the course of the oscillation time series, rms values characteristic for the whole record were calculated. The acceleration, speed, and deflection values were measured at points where significant horizontal and vertical oscillations resulting from traffic passage were expected. The measured maximum effective values of oscillation are summarised in Table 4. The values are obtained by integration in the range of 0 to 100 Hz. Any differences in the comparable measured oscillation speeds can be explained by the effect of different

loads during measurement when different traffic intensities occur. Table 4 shows the maximum values that may occur during normal road traffic.

The effective value of a time variable is obtained using the formula

$$v_{ef} = \sqrt{\lim_{T \to \infty} \frac{1}{T} \int_0^T (v(t))^2 dt}.$$

If the motion is harmonic (sinus), the maximum value of the quantity is found with

$$v_{\max} = v_{ef} \cdot \sqrt{2} = 1.41 v_{ef}.$$

**Table 4.** Characteristic values of individual measurements in acceleration, velocity, and deflection (RMS).

| | Response in a Point [RMS] | | | | | | | | | |
|---|---|---|---|---|---|---|---|---|---|---|
| | Velocity [mm/s$^1$] | | | | | | Deflection [mm] | | Acceleration [mm/s$^2$] | |
| **No. of Rec.** | 1 | | 2 | | 3 | | | | 1 | 2 |
| | z | x | z | x | z | x | z | x | z | z |
| | Street traffic | | | | | | | | | |
| 1 | 0.06012 | 0.06313 | 0.19176 | 0.22987 | 0.39743 | 0.12503 | 0.00350 | 0.00158 | 9.75610 | 26.07900 |
| 2 | 0.04860 | 0.04509 | 0.17434 | 0.18739 | 0.35291 | 0.09556 | 0.00314 | 0.00151 | 8.14330 | 23.46400 |
| 3 | 0.05110 | 0.05145 | 0.15941 | 0.27838 | 0.32168 | 0.11730 | 0.00281 | 0.00184 | 9.37850 | 22.18700 |
| 4 | 0.05579 | 0.05414 | 0.15374 | 0.23414 | 0.29965 | 0.10527 | 0.00275 | 0.00123 | 9.32040 | 21.80100 |
| 5 | 0.02261 | 0.04063 | 0.05824 | 0.08076 | 0.11924 | 0.04915 | 0.00121 | 0.00129 | 3.05160 | 7.83420 |
| 6 | 0.05284 | 0.06366 | 0.17375 | 0.26503 | 0.36874 | 0.10406 | 0.00329 | 0.00157 | 8.07680 | 22.94600 |
| 7 | 0.05687 | 0.06207 | 0.13641 | 0.20267 | 0.27011 | 0.10326 | 0.00238 | 0.00192 | 8.95600 | 19.18300 |
| 8 | 0.06048 | 0.06695 | 0.14606 | 0.24646 | 0.29635 | 0.10504 | 0.00278 | 0.00113 | 9.71150 | 20.32800 |
| 9 | 0.04147 | 0.04616 | 0.14043 | 0.18291 | 0.28687 | 0.08626 | 0.00254 | 0.00188 | 7.40440 | 19.38400 |
| 10 | 0.00480 | 0.02751 | 0.01626 | 0.02955 | 0.03883 | 0.03147 | 0.00066 | 0.00113 | 0.60418 | 1.80610 |
| 11 | 0.05952 | 0.06037 | 0.14246 | 0.21298 | 0.29291 | 0.10127 | 0.00253 | 0.00136 | 10.16800 | 20.83700 |
| 12 | 0.03869 | 0.05061 | 0.09165 | 0.16293 | 0.18645 | 0.09277 | 0.00186 | 0.00106 | 6.72200 | 13.51100 |
| 13 | 0.05437 | 0.05228 | 0.23018 | 0.22794 | 0.46664 | 0.11135 | 0.00413 | 0.00217 | 9.06750 | 30.82400 |
| 14 | 0.04906 | 0.04863 | 0.15614 | 0.18311 | 0.30854 | 0.09235 | 0.00272 | 0.00156 | 9.08660 | 22.12000 |
| 15 | 0.01148 | 0.03053 | 0.04048 | 0.04555 | 0.08603 | 0.04352 | 0.00090 | 0.00141 | 1.48770 | 5.67540 |
| 16 | 0.00417 | 0.02585 | 0.01106 | 0.05029 | 0.03246 | 0.08530 | 0.00054 | 0.00278 | 0.54676 | 1.70420 |
| 17 | 0.05698 | 0.05840 | 0.13207 | 0.20537 | 0.24795 | 0.09966 | 0.00239 | 0.00135 | 9.41030 | 18.74400 |
| 18 | 0.03044 | 0.04898 | 0.08060 | 0.12025 | 0.16473 | 0.07906 | 0.00156 | 0.00231 | 4.23280 | 11.19100 |
| 19 | 0.06574 | 0.06215 | 0.18553 | 0.24812 | 0.37103 | 0.12215 | 0.00342 | 0.00194 | 10.99500 | 25.96400 |
| 20 | 0.05071 | 0.05574 | 0.15407 | 0.17043 | 0.30785 | 0.08738 | 0.00279 | 0.00162 | 7.44600 | 20.76400 |
| 21 | 0.06626 | 0.06983 | 0.20016 | 0.23668 | 0.39737 | 0.13394 | 0.00348 | 0.00230 | 10.78300 | 27.95700 |
| 22 | 0.05822 | 0.05461 | 0.18642 | 0.29360 | 0.37949 | 0.12073 | 0.00326 | 0.00181 | 10.30200 | 26.35400 |
| 23 | 0.00596 | 0.02662 | 0.01441 | 0.03568 | 0.03207 | 0.05057 | 0.00040 | 0.00165 | 0.74264 | 2.06560 |
| 24 | 0.00491 | 0.02622 | 0.01355 | 0.03142 | 0.03316 | 0.03909 | 0.00047 | 0.00130 | 0.62152 | 2.10400 |
| 25 | 0.05919 | 0.04836 | 0.24439 | 0.26570 | 0.51258 | 0.39817 | 0.00464 | 0.01241 | 10.49400 | 33.54000 |
| | Shock load | | | | | | | | | |
| 26 | 0.00989 | 0.03376 | 0.30787 | 5.67980 | 0.24440 | 10.54600 | 0.00959 | 0.33906 | 0.79278 | 18.01200 |
| 27 | 0.01231 | 0.02508 | 0.20358 | 3.62420 | 0.27197 | 7.44010 | 0.00945 | 0.24262 | 1.50940 | 12.96900 |
| 28 | 0.04603 | 0.04386 | 0.12893 | 0.16853 | 0.25875 | 0.13993 | 0.00231 | 0.00409 | 7.17240 | 18.00100 |
| 29 | 0.01425 | 0.02746 | 0.13575 | 2.37780 | 0.63979 | 3.99050 | 0.02506 | 0.13805 | 1.93780 | 8.43150 |

Note: The maximum measurement results obtained during the tests are marked in grey.

The response to technical seismicity loads is usually assessed using the value of the effective speed of oscillation on the lowest floor or at the foundations of the building. These are called reference points. National standards give acceptable limits for these oscillations. The provisions of the Czech national standard ČSN 73 0040 admit that in other places on the

structure, the oscillation speeds may be higher. The values in Table 5 show that when excited by a tram crossing the intersection from West to East, the effective speed of oscillation on the pylon of the roof of the National Theatre building is the highest. Despite this, it is many times lower than the permissible values from the national ČSN 73 0040 standard. None of the measured values of the effective oscillation speed exceed the limit of 1.5 mms$^{-1}$ given by the ČSN 73 0040 standard.

**Table 5.** Amplitude of individual measurements in acceleration, velocity, and deflection.

| No. of Rec. | Velocity [mm/s$^1$] | | | | | | Defelection [mm] | | Acceleration [mm/s$^2$] | |
| | 1 | | 2 | | 3 | | | | 1 | 2 |
| | z | x | z | x | z | x | z | x | z | z |
|---|---|---|---|---|---|---|---|---|---|---|
| | | | | | **Street traffic** | | | | | |
| 1 | 0.27550 | 0.22123 | 0.74732 | 0.88624 | 1.49680 | 0.42697 | 0.01296 | 0.00505 | 44.05900 | 114.31000 |
| 2 | 0.18610 | 0.20494 | 0.62057 | 0.62168 | 1.17980 | 0.33838 | 0.01067 | 0.00452 | 33.42000 | 87.90800 |
| 3 | 0.24146 | 0.18397 | 0.59798 | 0.90583 | 1.12040 | 0.35117 | 0.00982 | 0.00608 | 38.56500 | 80.71500 |
| 4 | 0.19794 | 0.19189 | 0.46360 | 0.79628 | 0.89868 | 0.36466 | 0.00967 | 0.00400 | 37.03300 | 76.79900 |
| 5 | 0.08751 | 0.12943 | 0.22631 | 0.27103 | 0.53129 | 0.15891 | 0.00540 | 0.00384 | 11.11400 | 31.69900 |
| 6 | 0.21366 | 0.22515 | 0.64071 | 0.99242 | 1.24790 | 0.36586 | 0.01162 | 0.00441 | 30.97000 | 83.74100 |
| 7 | 0.22312 | 0.22202 | 0.47187 | 0.69147 | 0.86857 | 0.35325 | 0.00794 | 0.00548 | 34.68000 | 69.79300 |
| 8 | 0.27533 | 0.26591 | 0.56046 | 0.94398 | 1.12830 | 0.38450 | 0.00982 | 0.00393 | 48.25900 | 73.55500 |
| 9 | 0.15710 | 0.17336 | 0.47362 | 0.73061 | 0.95044 | 0.29646 | 0.00828 | 0.00464 | 30.47700 | 74.80000 |
| 10 | 0.01796 | 0.08123 | 0.11966 | 0.12728 | 0.12201 | 0.10512 | 0.00198 | 0.00367 | 2.65640 | 6.85250 |
| 11 | 0.23639 | 0.23856 | 0.55385 | 0.72714 | 1.04230 | 0.32458 | 0.00865 | 0.00399 | 38.47500 | 83.93800 |
| 12 | 0.15356 | 0.18138 | 0.37587 | 0.59500 | 0.70544 | 0.31870 | 0.00634 | 0.00326 | 31.26400 | 52.93500 |
| 13 | 0.21928 | 0.19607 | 0.81123 | 0.91937 | 1.62810 | 0.38866 | 0.01421 | 0.00572 | 41.50100 | 110.19000 |
| 14 | 0.20233 | 0.16715 | 0.54805 | 0.61969 | 1.06990 | 0.26420 | 0.00851 | 0.00414 | 37.88600 | 87.45900 |
| 15 | 0.04632 | 0.10846 | 0.13877 | 0.20922 | 0.33286 | 0.12359 | 0.00352 | 0.00431 | 6.31260 | 21.56700 |
| 16 | 0.01405 | 0.06697 | 0.04331 | 0.13427 | 0.11760 | 0.15755 | 0.00162 | 0.00515 | 1.93630 | 7.29540 |
| 17 | 0.23273 | 0.21990 | 0.45250 | 0.69074 | 0.80873 | 0.34246 | 0.00734 | 0.00401 | 42.43200 | 68.65400 |
| 18 | 0.12082 | 0.16000 | 0.34344 | 0.49640 | 0.58452 | 0.22713 | 0.00513 | 0.00604 | 17.30900 | 46.14300 |
| 19 | 0.23784 | 0.22892 | 0.61138 | 0.96369 | 1.22040 | 0.46838 | 0.01047 | 0.00487 | 41.75900 | 93.42100 |
| 20 | 0.19011 | 0.19266 | 0.47005 | 0.58252 | 0.88432 | 0.25976 | 0.00775 | 0.00485 | 25.99600 | 73.04700 |
| 21 | 0.26669 | 0.25391 | 0.70868 | 0.85744 | 1.26480 | 0.49415 | 0.01248 | 0.00650 | 36.37800 | 112.22000 |
| 22 | 0.21982 | 0.18971 | 0.60219 | 0.92357 | 1.37490 | 0.40847 | 0.01169 | 0.00499 | 40.45400 | 93.65400 |
| 23 | 0.02062 | 0.07200 | 0.04664 | 0.10972 | 0.10833 | 0.11672 | 0.00136 | 0.00394 | 2.72280 | 6.93760 |
| 24 | 0.01776 | 0.06716 | 0.04563 | 0.11359 | 0.11044 | 0.08844 | 0.00147 | 0.00299 | 2.17550 | 8.19280 |
| 25 | 0.21883 | 0.19149 | 0.72082 | 1.04510 | 1.44970 | 1.73000 | 0.01571 | 0.06493 | 40.45000 | 109.99000 |
| | | | | | **Shock load** | | | | | |
| 26 | 0.02281 | 0.07369 | 0.62688 | 10.96500 | 0.54518 | 20.55300 | 0.01925 | 0.65392 | 4.78180 | 37.87300 |
| 27 | 0.05109 | 0.06916 | 0.49448 | 7.99920 | 0.71822 | 16.18200 | 0.02494 | 0.53874 | 5.21830 | 39.84200 |
| 28 | 0.20480 | 0.13551 | 0.47439 | 0.62417 | 0.99862 | 0.37091 | 0.00846 | 0.00892 | 29.36100 | 77.30400 |
| 29 | 0.06439 | 0.10829 | 0.31212 | 4.94160 | 2.63100 | 8.43620 | 0.09511 | 0.31036 | 7.00100 | 23.16800 |

Note: The maximum measurement results obtained during the tests are marked in grey.

For the measurement and evaluation of the vibration level according to the standard, the velocity of the vibration is recommended. The response to technical seismic loading is generally measured and assessed by the value of the effective vibration velocity at the lowest floor or at the foundation of the building; these locations are called reference points. However, at other locations in the structure, the observed vibration velocities may be greater than at the reference point. The dynamic response due to technical seismicity, with the exception of the response due to blasting in terms of bearing capacity, does not need to be analysed further if the effective vibration velocity at the reference point does not exceed the limits given in the standard. The measurement and analysis of deflection and acceleration values are used only for information or for comparison with the calculation.

Evaluation of the measurements revealed that the structure of the National Theatre is not subjected to any unacceptable mechanical vibrations.

The measured values of dynamic response for everyday operating conditions show that there is no oscillation of the object that could adversely affect the triga sculpture. The measured response values are minuscule. Since the response is small, there is no need to prescribe any anti-vibration measures. The dynamic response caused by passing cars does not need to be further measured and analysed. However, the statues may vibrate in windy weather. The resulting deviations in the horizontal direction perpendicular to the horse's axis may then be significantly higher than with the effects of traffic (see Table 5).

## 3. Assessment of a Renaissance Brick Vault Ceiling

The study load test described in this paragraph was prompted by the need to know and evaluate the contribution of the load on the central part of the exhibition space of the first floor of the renaissance chateau Hvězda in Prague to the deformation of the vault above the ground floor and the possible development of cracks. A request was also made to comment on the vault damage and to formulate recommendations for further measures to prevent their progressive development and possible serious consequences. The report is based on deflection measurements and visual observations of cracks in the vaults during on-site investigations (Figure 12).

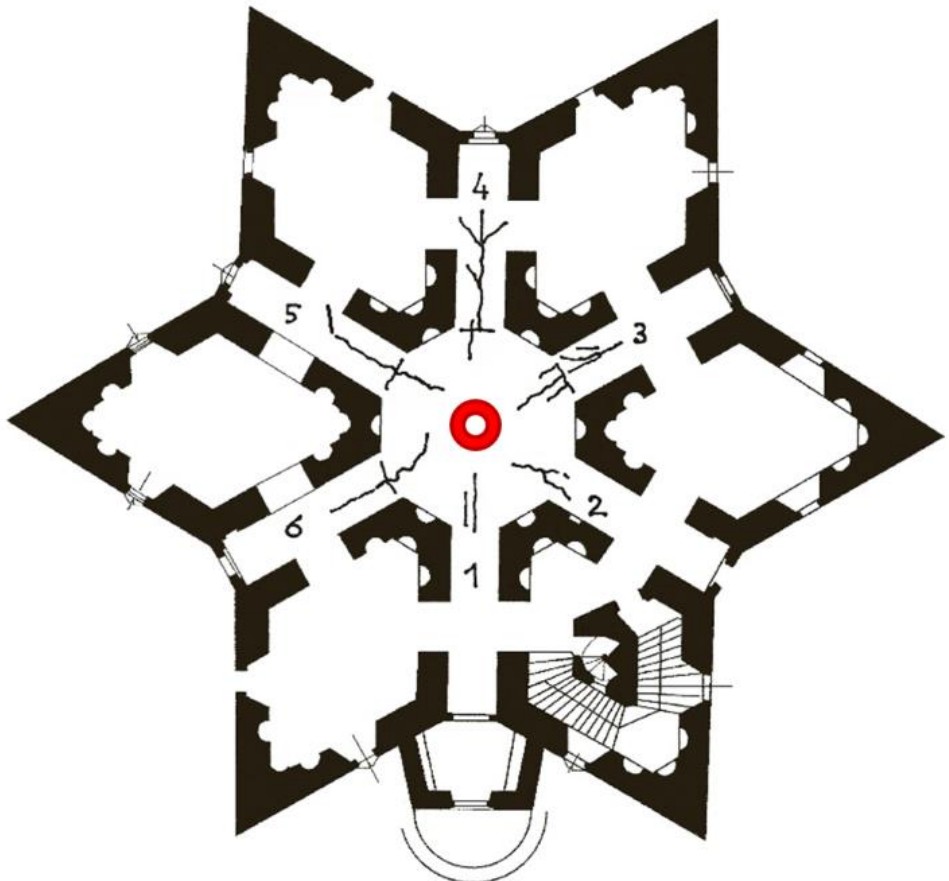

**Figure 12.** Ground plan of the Hvězda Summer chateau with marked faults in the vaults and the position of loading and deflection measurement on the central vault.

Construction of the Hvězda Summer chateau in Prague (Czech Republic) took place during the years 1555–1580. The building was restored at the turn of the forties and fifties of the twentieth century and adapted for the purposes of the "Alois Jirásek" Museum, which is used for permanent as well as temporary exhibitions.

### 3.1. Load Test Arrangement and Results

During the study load test, the vault was loaded by a group of people of known weight in order to model a load typical of exhibition visits. In the middle of the central space, there is a chipboard construction (Figure 13), adapted for visitors to sit and listen to the recordings accompanying the bands projected onto screens in radially spreading corridors. Therefore, the live load during the exhibition is well localised on the vault.

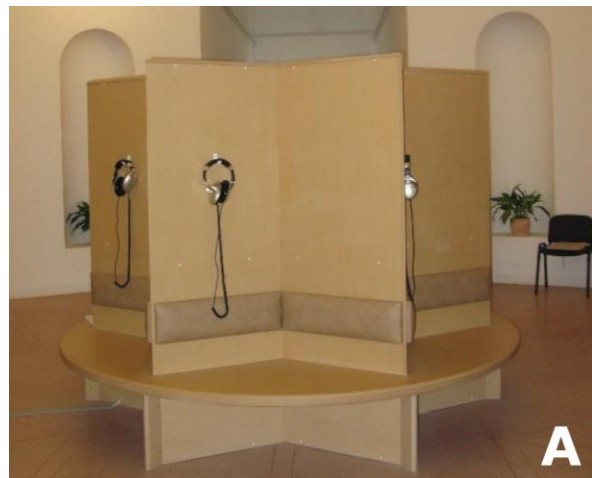 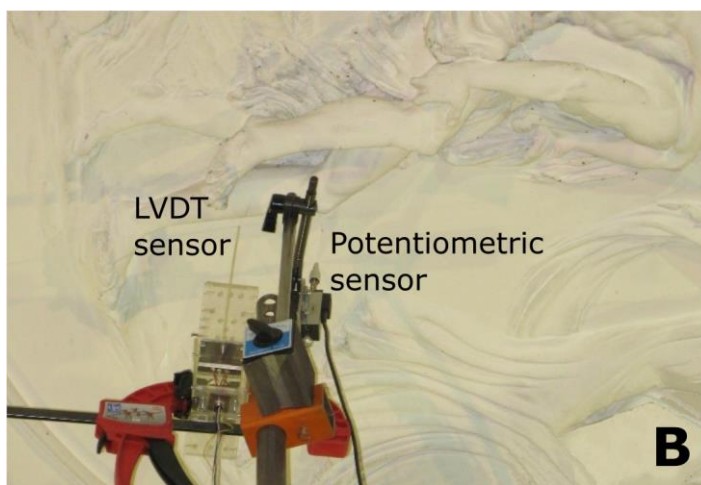

**Figure 13.** Seating structure for listening to recordings accompanying the exhibition (**A**). During the test, the loading persons with a total weight of 552 kg (first series) and 867 kg (second series) sat in the seats. Measurement of deflections using an LVDT sensor and a potentiometric sensor (**B**).

Three load arrangements were used during the tests. During the basic load condition, a group of people was situated on the above-mentioned structure as close as possible to the centre of the vault. In the second load case, a group of people stood surrounding the structure at a distance of about 0.5 m from the seats, and in the third load case, the vault was loaded with a dynamic impact by the jumping of people. The test was repeated twice: the first time with a group of persons with a total mass of 552 kg, the second time with a total of 867 kg.

Measurements on the vault with its valuable stucco decoration did not allow for the attachment of sensors that would be needed to monitor movement in the crack. Therefore, only the deflection was measured using sensors that touch the surface of the vault. A very sensitive linear variable differential transformer (LVDT) sensor with a range of 1 mm and a linear potentiometric displacement sensor were used for inspection (Figure 13). The stands with sensors were placed on a light portable duralumin scaffolding (Figure 14). Before the test, vault defects were identified, manifested by a system of cracks, and their behaviour was visually monitored during the test with photographic recording. The basic system of cracks is indicated in Figure 12 and partly corresponds to the findings recorded during an inspection in the year 1991. At that time, no detailed documentation was made, but the historical report does not mention the oblique cracks in the barrel vaults of some corridors, which therefore possibly developed later. Without a more detailed analysis or probes or monitoring, it is difficult to evaluate their origin and development. On the surface of the vaults, additional cracks are noticeable that are related to the technology of stucco decoration and the peeling of thin layers of paint, which can be observed even by visitors.

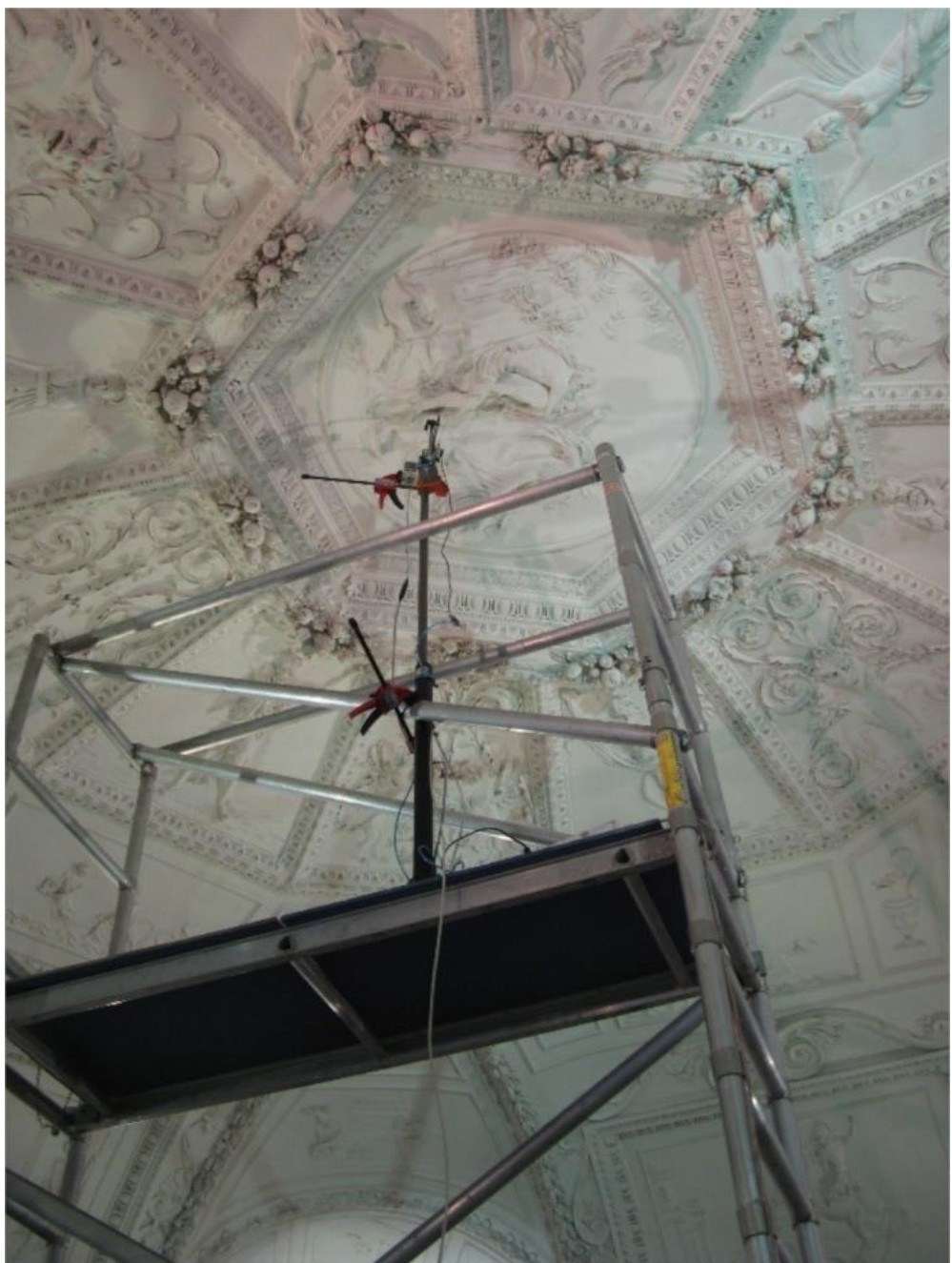

**Figure 14.** Placement of the deflection sensors on a light scaffolding. The stucco decoration influenced the choice of location for the point of contact of the sensors with the vault, and therefore the measurement could not be carried out exactly in the middle of the vault.

The record of the vault deflections under the various load cases is shown in Figures 15 and 16. In the first series of measurements (Figure 15), the deflection reached 19.1 μm in the first load test and 10.47 μm in the subsequent load test. The graph shows very well that the deflection decreases very quickly when the loading persons move away from the centre of the vault. Dynamic loads at a distance of 0.5 m from the seats produced approximately the same effects as those of the persons seated on the central seating structure.

In the second series of measurements (Figure 16), the deflection in the centre of the vault reached 31.5 μm in the first load test (persons sitting on the seats) and approximately 19.56 μm in the subsequent load test. The ratio of static to dynamic effects is similar to that in the first series.

In the deflection measurement using the potentiometric sensor during the second load series, a deflection of approximately 41 μm was recorded. The difference, compared to the LVDT sensor measurement, is due to the lower sensitivity and accuracy of the potentiometric sensor. Nevertheless, at such low values, the measurement compliance is reasonably good.

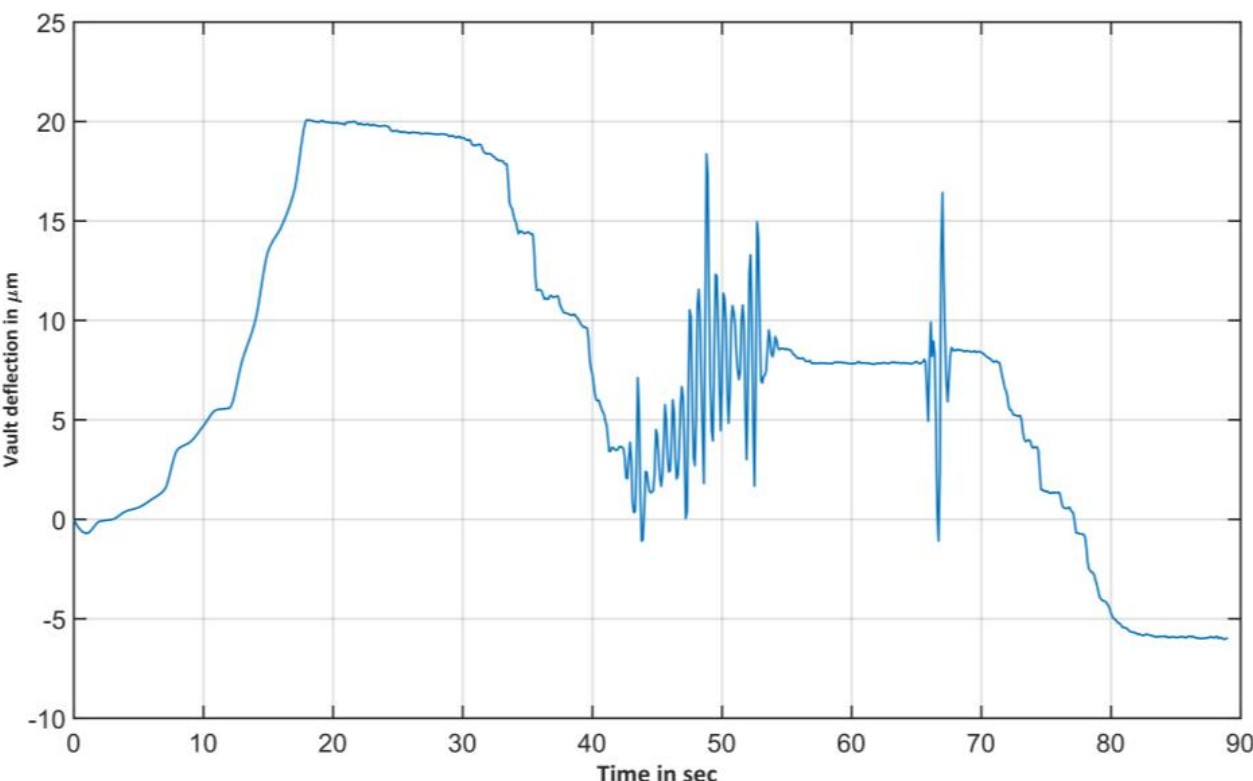

**Figure 15.** Measured values of deflections in the centre of the vault when the load group was seated (left horizontal part of the graph) and when they moved away (right horizontal part of the graph). The central horizontal part of the graph (top left) corresponds to the load of persons standing still at a distance of about 0.5 m from the seats. The left group of oscillations in the right part of the graph corresponds to the cyclic rocking of persons in a circle 0.5 m from the seats and the right oscillation to the jump of persons in a circle 0.5 m from the seats.

### 3.2. Load Test Findings

The load tests and the visual inspection yielded several results. The vault, due to its shallow nature, is very sensitive to loads in the middle of its span, approximately in the area given by a circle with a radius of the installed seat increased by about 0.5 m. Further, the growth of deflections when changing the position of the load is nonlinear and corresponds to the typical course for vaults of a similar type. Under loading, there were no visible changes in the crack system. However, the movements in the cracks are very small due to the size of the deflections, and their monitoring would require the placement of special sensors. It can be expected that the loading of the vault contributes to the cracks' propagation at the connection of the lunettes to the corridors due to a different stiffness of the connected elements. The bench installed in the middle of the room improperly loads the vault by attracting visitors to gather in the least suitable area in terms of the resulting strain on the structure. Although this condition undoubtedly contributes to the gradual deterioration of the cracks, there is no imminent accident or significantly progressive opening of the cracks.

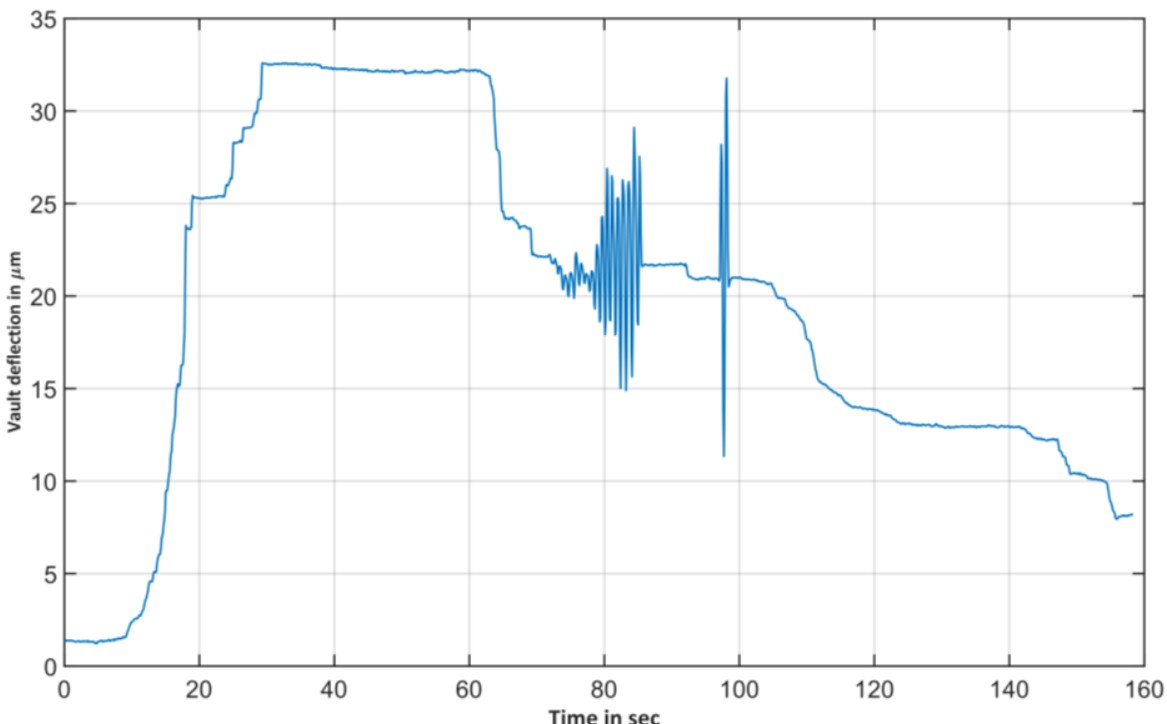

**Figure 16.** Measured values of deflections in the centre of the vault in the second series of measurements under the load group weighing 867 kg. The meaning of the individual parts of the graph is the same as in Figure 15, with the only difference being that there is a longer part showing the deflection of the vault completely unloaded (the left horizontal part of the graph).

## 4. Conclusions

The load tests performed provided the necessary knowledge to answer the questions posed about the behaviour of the tested sculpture and structure.

In the case of the bronze sculpture, the proportion of load transfer by the individual elements of the system consisting of the bronze shell and the iron skeleton was determined. The response to stress caused by technical seismicity was evaluated using the effective vibration velocity (RMS). Assessment of the measurement indicates that the construction is not subjected to unacceptable mechanical vibrations. The values measured in the horizontal and vertical directions at the selected points are below the standard allowable limits. The tests also showed that wind loads, which were simulated by a horizontal impulse load, can be significant and repeat regularly, which may cause local fatigue damage.

In the case of the floor supported by a masonry vault, it was concluded that its static and dynamic loading by visitors of the temporary exhibitions would not cause dangerous deflections or failures.

**Author Contributions:** Conceptualization, M.D.; methodology, M.D. and S.U.; software, S.U.; validation, M.D. and S.U.; formal analysis, M.D. and S.U.; investigation, M.D. and S.U.; resources, M.D. and S.U.; data curation, M.D. and S.U.; writing—original draft preparation, M.D. and S.U.; writing—review and editing, M.D. and S.U.; visualization, M.D.; supervision, M.D.; project administration, M.D. and S.U.; funding acquisition, M.D. All authors have read and agreed to the published version of the manuscript.

**Funding:** This research received no external funding.

**Informed Consent Statement:** Not applicable.

**Data Availability Statement:** The research data are available on the request from the authors.

**Acknowledgments:** This paper is partly based on the results of research supported by the institutional project RVO 68378297 and by the Strategy AV21 programme. The authors thank Ing. Pavel Beneš for careful preparing the figures, and Marek Eisler for English correction.

**Conflicts of Interest:** The authors declare no conflict of interest.

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
