# Peer review of "Load Testing of Cultural Heritage Structures and Sculptures: Unconventional Methods for Assessing Safety"

_heritage, doi:10.3390/heritage6070292_

Round 1
Reviewer 1 Report
The paper provides exciting research for the readers of the journal, however, there are some issues which the authors must address:
1. The literature review should be enhanced by including other similar studies.
2. The results and data need to be further discussed.
3. The word RMS and parentheses have been overused, please revise them.
4. Figures, particularly figures 15 and 16, should be improved.
The paper needs to be proofread.
Author Response
Thank you very much for the careful reading and useful suggestions or corrections.
The paper provides exciting research for the readers of the journal, however, there are some issues which the authors must address:
- The literature review should be enhanced by including other similar studies.
Testing of cultural heritage objects or structures is not frequently performed and published. The studies are typically available in the grey literature archives. Selected examples of papers on proof load tests of existing structures even though not of the cultural heritage value are added.
- The results and data need to be further discussed.
The data and results have been further discussed in the text.
- The word RMS and parentheses have been overused, please revise them.
Thank you for the comment. The overused (RMS) have been deleted.
- Figures, particularly figures 15 and 16, should be improved.
Figures 4, 5, 9, 13, 15, 16 were improved as well as their captions. An explanatory window was added to the picture 9 which was lost in the manuscript.
Reviewer 2 Report
In tables 4 and 5 the grey enhancement of the maximum results is lacking.
Author Response
In tables 4 and 5 the grey enhancement of the maximum results is lacking.
Thank you very much for your notice. It has been corrected.
Reviewer 3 Report
Comments to the Author(s)
This manuscript presents Load testing of cultural heritage structures and sculptures: Un-conventional Methods for Assessing Safety. This paper contains a good effort related to cultural heritage structures and sculptures because they are used to ensure the health of these structures. This paper contains good material of interest to the cultural heritage community. In general, the manuscript is well organized. It has clear and simple text to read. However, the authors are suggested to improve the paper by resolving the following issues (All the answers should be included in the manuscript):
1. Please enhance the introduction with at least 10 recent references.
2. Page 6, line 179, please write the full words of “MTS” and “LUKAS”.
3. Fig. 4, please label the items in the images.
4. Fig. 5, please identify the strain gauge marked “CH, PV… etc.”.
5. Fig. 10, Please label plots as a, b. For the caption of Fig 10, please explain Fig. 10a and Fig. 10b
6. Fig. 11, there are some peaks (resonance peaks). Please discuss these peaks.
7. Page 13, line 290, you mention “None of the measured values of the effective oscillation speed (RMS) exceed the limit of 1.5 mms-1 given by the ÄŒSN 73 0040 standard”. What is about other measurements (deflection, acceleration)?
8. Fig. 13, please label these two images (a, b) instead of saying left or right image. Also, label the items of these two images.

Author Response
This manuscript presents Load testing of cultural heritage structures and sculptures: Un-conventional Methods for Assessing Safety. This paper contains a good effort related to cultural heritage structures and sculptures because they are used to ensure the health of these structures. This paper contains good material of interest to the cultural heritage community. In general, the manuscript is well organized. It has clear and simple text to read. However, the authors are suggested to improve the paper by resolving the following issues (All the answers should be included in the manuscript):
Thank you very much for the careful reading and useful suggestions or corrections.
- Please enhance the introduction with at least 10 recent references.
Testing of cultural heritage objects or structures is not frequently performed and published. The studies are typically available in the grey literature archives. Selected examples of papers on proof load tests of existing structures even though not of the cultural heritage value are added.
- Page 6, line 179, please write the full words of “MTS” and “LUKAS”.
MTS and LUKAS are Trade Marks which are generally well known to professionals. Nevertheless, we added in parentheses MTS in full words (Material Test Systems). LUKAS is the surname of the load cell producer and this is not possible to write in full words. We added quotation marks.
- Fig. 4, please label the items in the images.
Thank you for your suggestion. The pictures were labeled and an explanatory window added to the picture 9.
- Fig. 5, please identify the strain gauge marked “CH, PV… etc.”.
There was explained in the text that the CH in the picture means “channel” of the measurement device. There was further explained marking in the strain gages on the reinforcing steel frame to PV with orientation of its side to the North (S) or to the South (J).
- Fig. 10, Please label plots as a, b. For the caption of Fig 10, please explain Fig. 10a and Fig. 10b
The captions have been improved. …during the lifting and the lowering of the sculpture.
- Fig. 11, there are some peaks (resonance peaks). Please discuss these peaks.
The frequency components of the time record from the massive building where the statue is located are shown in the pictures above. The frequency composition in the vertical and horizontal directions corresponds to the wide band spectrum typical for dynamic loading of the induced traffic. The figures in the middle and bottom correspond to the frequency of the statue in the vertical and horizontal directions. In these cases, the spectrum peaks show the natural frequency of the statue, which can be amplified at the resonance excitation frequencies.
- Page 13, line 290, you mention “None of the measured values of the effective oscillation speed (RMS) exceed the limit of 1.5 mms-1 given by the ÄŒSN 73 0040 standard”. What is about other measurements (deflection, acceleration)?
For the measurement and evaluation of the vibration level according to the standard, the velocity of the vibration is recommended. The response to technical seismic loading is generally measured and assessed by the value of the effective vibration velocity at the lowest floor or at the foundation of the building; these locations are called reference points. However, at other locations in the structure, the observed vibration velocities may be greater than at the reference point. The dynamic response due to technical seismicity, with the exception of the response due to blasting in terms of bearing capacity, does not need to be analysed further if the effective vibration velocity at the reference point does not exceed the limits given in the standard. The measurement and analysis of deflection and acceleration values are used only for information or for comparison with the calculation.
- Fig. 13, please label these two images (a, b) instead of saying left or right image. Also, label the items of these two images.
The images were improved with the required labeling.
Round 2
Reviewer 1 Report
The paper has been revised properly and can be considered for publication.